# Scalable Multi-Domain Adaptation of Language Models using Modular Experts

## Abstract

Domain-specific adaptation is critical to maximizing the performance of pre-trained language models (PLMs) on one or multiple targeted tasks, especially under resource-constrained use cases, such as edge devices. However, existing methods often struggle to balance domain-specific performance, retention of general knowledge, and efficiency for training and inference. To address these challenges, we propose Modular Domain Experts (MoDE). MoDE is a mixture-of-experts architecture that augments a general PLMs with modular, domain-specialized experts. These experts are trained independently and composed together via a lightweight training process. In contrast to standard low-rank adaptation methods, each MoDE expert consists of several transformer layers which scale better with more training examples and larger parameter counts. Our evaluation demonstrates that MoDE achieves comparable target performances to full parameter fine-tuning while achieving 1.65% better retention performance. Moreover, MoDE's architecture enables flexible sharding configurations and improves training speeds by up to 38% over state-of-the-art distributed training configurations.

## 1 Introduction

Recent advances in large-scale Pre-trained Language Models (PLMs) have showcased impressive generalization capabilities (Brown et al., 2020; Chowdhery et al., 2023; Anil et al., 2023; Team et al., 2023). However, when applied to specialized domains such as medical, legal, or financial sectors, these general-purpose models often require further fine-tuning to maximize performance on target domains (Huang et al., 2023; Li et al., 2023; Singhal et al., 2023).

A straightforward approach to domain adaptation is full-parameter fine-tuning, where the entire model is further trained on domain-specific data (Houlsby et al., 2019; Bapna et al., 2019). While this method provides strong performance on target domains, full-parameter fine-tuning may lead to *catastrophic forgetting* where the model loses previously learned capabilities by overfitting to the target domain (Goodfellow et al., 2013; Kirkpatrick et al., 2017). Additionally, this method is memory-intensive to serve in multi-domain settings as each domain has a unique set of parameters, incurring a significant parameter loading overhead when switching between domains (Hu et al., 2021). In such cases, the cost of frequent "context switches" significantly impacts performance, making full-parameter fine-tuning impractical for scalable, efficient deployment (Dhar et al., 2024).

To address the issues of forgetting and memory-efficiency, parameter-efficient fine-tuning methods have been proposed, such as adapter modules (Houlsby et al., 2019), LoRA (Hu et al., 2021), and CoDA (Lei et al., 2023). These methods introduce a small number of trainable parameters and keep the original model frozen during training. Through targeted parameter updates, these approaches are both computationally efficient and effective at retaining prior knowledge (Biderman et al., 2024). Despite these benefits, parameter-efficient methods are limited in their expressive potential and struggle to scale effectively across large domains and datasets (Niederfahrenhorst et al., 2023).

In this work, we propose Modular Domain Experts (MoDE), a scalable method for multi-domain adaptation. MoDE is inspired by the Mixture of Experts (MoE) approach (detailed in Section 2), and extends PLMs by introducing modular, composable domain-specialized experts. In MoDE, each expert consists of several transformer layers that operate in parallel to the "backbone" PLM. During training, the backbone remains frozen, and each expert is independently trained for its respective domain. For multi-domain deployment, MoDE combines multiple experts with different

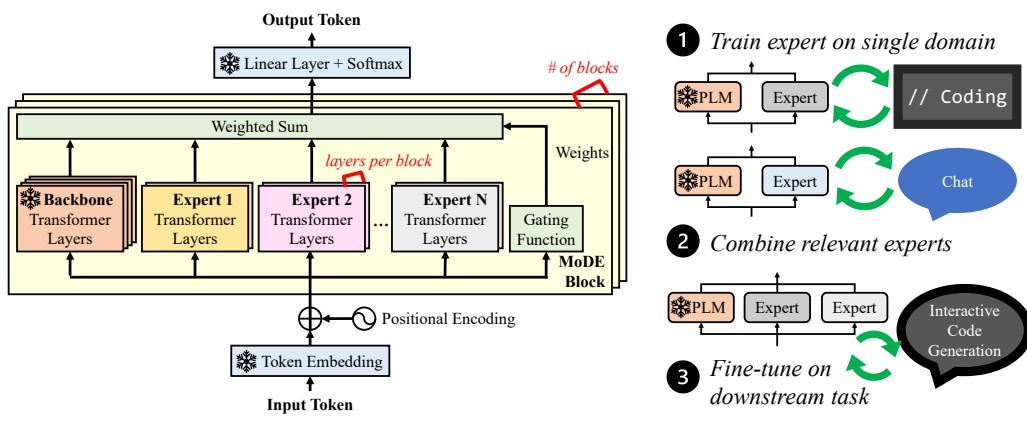

(a) Model architecture.  (b) Training procedure.

Figure 1: **MoDE overview.** MoDE models are divided into blocks, each containing transformer layers from the backbone or an expert (Figure 1a). Backbone and expert blocks operate on the same inputs. The model takes a linear combination of their outputs, where the weights are determined by a gating function. Figure 1b outlines the training process: **1** Experts are trained independently on specific domains, while the backbone's parameters remain unchanged. **2** For a multi-domain task, experts are modularly composed to enhance the model's performance. Here, the code and chat experts are combined to improve the performance of an interactive coding assistant. **3** A lightweight fine-tuning process updates the experts and the gating function to improve performance on the target task.

specializations by placing them in parallel to the backbone PLM, allowing for strong performance across diverse domains. A lightweight fine-tuning steps teaches the experts to collaborate effectively to provide strong performance across all target domains. We further take advantage of parallelism in MoDE's architecture to enable new sharding strategies that improve training efficiency, and provide privacy benefits during inference.

In summary, we make the following contributions:

- We introduce the novel Modular Domain Experts (MoDE) architecture designed to address the scalability, catastrophic forgetting, and memory constraints in multi-domain adaptation.
- We evaluate the performance of MoDE and demonstrate that it outperforms LoRA by 1.4% and full parameter fine-tuning by 0.6% on a multi-domain task.
- We analyze the training efficiency of MoDE, showing better scalability compared to LoRA as the number of training examples and parameters increases.
- We demonstrate how MoDE's design enables flexible sharding strategies that accelerate training, achieving up to 38% speedup over standard distributed training configurations.

## 2 RELATED WORK

**Domain-Adaptive Pre-training.** MoDE is designed for scenarios where large domain-specific unlabeled datasets are available, a process often referred to as "domain-adaptive pre-training", "continual learning", "continued pre-training", or "further pre-training" (Shi et al., 2024; Azerbayev et al., 2023; Colombo et al., 2024; Agarwal et al., 2024). Beyond single-domain adaptation, recent research has expanded into multi-domain adaptation, which presents additional challenges (Saunders, 2022; Wu et al., 2024). To the best of our knowledge, the most relevant works are parameter-efficient fine-tuning methods, which are described in detail in the next paragraph.

**Parameter-Efficient Fine-Tuning.** Significant progress has been made in parameter-efficient fine-tuning methods, which update only a small subset of model parameters (Houlsby et al., 2019; He et al., 2021). Techniques such as LoRA (Hu et al., 2021) and QLoRA (Dettmers et al., 2023) enhance language model performance by introducing trainable low-rank decomposition matrices. However, these low-rank approaches often fall short when adapting to domains that require high expressiveness, such as mathematical reasoning or coding (Biderman et al., 2024; Niederfahrenhorst et al., 2023).

**Mixtures of Experts** (MoE) architectures are promising for expanding the capacity of language models while keeping the computational cost of training constant. MoE models learn a routing function that selectively activates a subset of experts, enabling sparse computation (Du et al., 2022; Zhou et al., 2022). Unlike MoDE, MoE models are primarily used to enhance pre-training performance (Dai et al., 2024; Team, 2024; xAI, 2024). Recent approaches extend MoE concepts to improve adaptation to new domains, such as life-long learning with distribution-specialized experts (Chen et al., 2023), conditional adapters (Lei et al., 2023), AdaMix (Wang et al., 2022), and MixPHM (Jiang & Zheng, 2023). These approaches modify PLMs at a fine granularity (e.g., by modifying the feed-forward network in the transformer layers). In contrast, MoDE applies MoE at the transformer-level, offering a scalable and expressive solution for multi-domain adaptation.

Furthermore, MoE-inspired approaches have shown potential in selecting domain-specific adapters by routing input sequences, which enhances model performance across diverse domains while alleviating catastrophic forgetting (Feng et al., 2024). In this work, we focus on token-level routing, with sequence-level routing as a natural extension for future research.

**Distributed Training.** Scaling PLMs to billions of parameters requires partitioning parameters and models and training data across multiple processors (Dean et al., 2012; Li et al., 2014; Barham et al., 2022). Common sharding strategies, including *model parallelism* and *data parallelism*, evenly divide inputs and parameters across accelerators (Rajbhandari et al., 2020; Lepikhin et al., 2020; Alabed et al., 2024). To address memory overheads and communication bottlenecks, researchers have developed specialized tools to optimize the sharding configuration of a model for the available processors (Lepikhin et al., 2020; Alabed et al., 2024).

Typically, distributed training and serving frameworks implement the *Single Program, Multiple Data* (SPMD) model of computation which enables building models and training programs as if developing for a single processor with a large pool of memory. However, SPMD has two key limitations: (*i*) all operations (i.e., layers of a model) execute sequentially, precluding the execution of different operations in parallel on different processors, and (*ii*) all operations must use all devices, which may be inefficient (e.g., by incurring significant communication overhead). In contrast, the *Multiple Program, Multiple Data (MPMD)* computational model has the ability to run multiple programs in parallel on different devices, enabling fine-grained control over parallelism which has the potential for more efficient execution compared to SPMD (Zheng et al., 2022). We exploit parallelism in MoDE's structure to implement an MPMD-enabled sharding strategy that accelerates training by up to 38% over SPMD model parallelism.

## 3 METHOD

Our proposed model architecture augments PLMs with modular, domain-specialized experts and aims to achieve the following design goals:

- **Domain specialization.** Experts must provide strong performance and high expressiveness on their target domains. Moreover, composing multiple experts with a PLM should result in strong performance on each of the target domains.
- **Capability retention.** The model must preserve the generalist capabilities of the PLM and avoid catastrophic forgetting which may cause performance regressions on other tasks.
- **Efficient training and inference.** The model architecture must be efficient to train on large clusters, and provide advantages for inference on edge devices.

### 3.1 MODEL DESIGN

To meet these design goals, we decompose the language model into multiple MoDE *blocks* (see Figure 1a). Each $i$th *block* contains: (*i*) a few backbone transformer layers, $f_{bb}^i(\cdot)$, which are initialized from PLMs, and (*ii*) a set of $N$ experts where each expert has the same (small) number of transformer layers, and the $i$th block of the $j$th expert is given by $f_e^{i,j}(\cdot)$. All backbone and expert layers run in parallel, with their outputs combined as a weighted sum. The weights for each component are determined by a lightweight gating function, $g^i(\cdot)$.

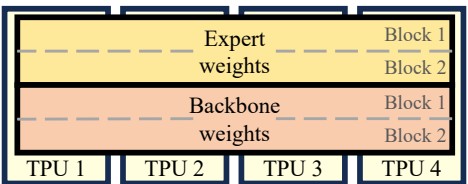 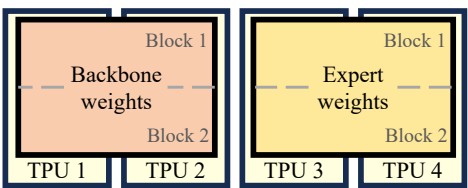

(a) A model-parallel partitioning in which an entire MoDE model is split across all TPUs.

(b) An flexible sharding configuration enabled by the MoDE architecture and MPMD.

Figure 2: **Example SPMD and MPMD sharding configurations.** While the model parallel sharding configuration enabled by SPMD evenly distributes all weights across all TPUs, the provided MPMD sharding configuration executes the backbone and the expert on 2 different meshes consisting of 2 TPUs each which may reduce communication overheads.

For each $i$-th block, the input $x \in \mathbb{R}^d$ and the output $y \in \mathbb{R}^d$ are defined as:

$$y = \alpha_{\text{bb}} \, f_{\text{bb}}^i(x) + \sum_{j=1}^{N} \alpha_{\text{e}}^j \, f_{\text{e}}^{i,j}(x) \tag{1}$$

$$\text{where} \quad [\alpha_{\text{bb}}, \alpha_{\text{e}}^1, \alpha_{\text{e}}^2, \cdots, \alpha_{\text{e}}^N] = g^i(x)$$

**Backbone Model.** The backbone transformer layers, $f_{\text{bb}}^i(x)$ are derived from PLMs. In each MoDE block, the number of backbone layers is determined by dividing the total number of layers in the original PLM by the total number of MoDE blocks. The number of MoDE blocks is an important configuration hyperparameter: more blocks enables more frequent synchronization between the backbone and expert layers which provides greater flexibility. In Table 3, we conduct an ablation to assess the impact of the number of blocks on accuracy. In MoDE, the backbone layers are initialized from PLMs and remain frozen during training to mitigate the issue of catastrophic forgetting.

**Experts.** Each expert, $f_{\text{e}}^{i,j}(x)$ consists of several transformer layers, and the number of transformer layers impacts the expert's performance. Table 3 indicates that increasing the number of expert layers improves performance on the target domain at the cost of higher computational cost. We use transformer layers for experts for two reasons: ($i$) transformer blocks scale more efficiently than finer-grained designs, such as modifications within attention matrices, and ($ii$) they simplify deployment, as there are no modifications to the transformer architecture itself. In contrast, fine-grained methods often face deployment challenges (Yi et al., 2023).

**Gating Function.** The outputs from the backbone and expert layers are combined through the gating layer, $g^i(x)$. In this work, we use a simple token-level gating function. This function consists of linear layer followed by a softmax for normalization. The gating function takes $x$ as input and outputs a vector of length $N + 1$ as follows:

$$g^i(x) = \texttt{softmax}(\texttt{Linear}(x)) \tag{2}$$

The simplicity of MoDE's gating function enables efficient composition of multiple experts, as shown in Table 1. However, future work could explore more advanced designs, such as sparse gating and sequence routing, to further optimize memory usage and computational efficiency.

## 3.2 TRAINING PROCEDURE

MoDE has a two stage training procedure (Figure 1b): ($i$) train a single expert independently on each domain, and ($ii$) compose different experts into a single model to improve multi-domain performance.

**Single Modular Expert Training.** For each domain, we independently train an MoDE model with a single expert. During training, only the expert layers and gating layers are updated, while the backbone layers remain frozen. Freezing the backbone not only reduces the computational cost of backpropagation but also helps mitigate catastrophic forgetting.

**Composing Experts.** After training the modular experts on each domain, we can compose them into a single MoDE model for multi-domain adaptation. While the experts contain domain-specific

knowledge, their outputs can interfere when combined, as they are pre-trained individually. To address this, we apply a lightweight fine-tuning step using data from all domains to learn the optimal weights for the gating function. Although freezing the experts requires fewer training examples during composition Appendix C, we have demonstrated that unfreezing the experts during this step further enhances multi-domain adaptation performance Table 1.

## 3.3 PARALLELIZING BLOCK EXECUTION

We exploit parallelism in MoDE's structure to enable flexible sharding configurations supported by the MPMD model of computation. These sharding configurations improve training efficiency due to:

1. The ability to execute the backbone and experts on distinct, smaller sets of devices which reduces communication overheads (Figure 2).
2. The ability to configure the backbone and experts independently to maximize overall efficiency, when their number of parameters are different.

While MPMD has the potential to reduce communication overheads by running blocks on fewer devices, the merges in the MoDE model architecture are points of synchronization that require communication across all devices. To determine whether MPMD is beneficial for training, we evaluate whether MPMD configurations can afford the cost of resharding between meshes in Section 4.4.

While MPMD has the potential to reduce communication overheads by executing blocks on smaller sets of devices, the linear combination of the outputs from the backbone and expert blocks requires are points of synchronization that require communication across all devices. To assess the effectiveness of MPMD in training, we evaluate if the benefits of MPMD configurations outweigh the costs associated with resharding between device meshes, as discussed in Section 4.4.

Moreover, MoDE's architecture enables several benefits during inference for privacy and performance. By using MPMD to run MoDE with the backbone and expert parameters on different devices, expert parameters for sensitive domains (e.g., personal information, medical data, etc.) can remain private (e.g., by executing on a user's phone). To switch to another MoDE model adapted to different task, only the new expert weights need to be loaded into memory because the backbone PLM's weights are shared. This reduces the overhead of loading model weights and works well with existing techniques developer to serve many adapters concurrently, such as SLoRA (Sheng et al., 2024).

## 4 EXPERIMENTS

### 4.1 EXPERIMENT SETUP

**PLM Configuration.** Our PLM model consists of 1.58 billion parameters distributed across 18 transformer layers, comparable to smaller open-source models such as Gemma (Team et al., 2024), Phi (Gunasekar et al., 2023), and Llama 3.2 (Meta, 2024). The model is pre-trained on a high-quality dataset that spans a diverse range of natural language use cases, similar to GPT-3 (Brown et al., 2020), GLaM (Du et al., 2022), and LLaMA (Touvron et al., 2023).

**Multi-Domain Datasets.** We prepare two target domain datasets, *Code* and *Math*, to evaluate multi-domain adaptation, and one retention dataset, *English*, to measure catastrophic forgetting:

- *Code* consists of code samples retrieved from real-world applications.
- *Math* tests the model's mathematical reasoning capabilities and is similar to the GSM (Cobbe et al., 2021) and MATH (Hendrycks et al., 2021) datasets.
- *English* contains literature texts with a distribution distinct from *Code* and *Math*.

Additionally, we create a mixed dataset, *Math + Code*, for model training, which contains an equal number of samples from the *Math* and *Code* datasets.

**Evaluation Metric.** As our focus is on domain-adaptive pre-training with large amounts of unlabeled data, we adopt next-token prediction accuracy as the primary metric to assess the effectiveness of our methods. This metric has been shown to correlate strongly with downstream performance (Kaplan et al., 2020; Hoffmann et al., 2022). In addition, since the evaluation spans multiple domains and includes a retention dataset, we use the average performance across all datasets to rank the models.

| Method | | Accuracy | | | |
|---|---|---|---|---|---|
| Method | Parameters | Math | Code | English | Average |
| Full-parameter Fine-tuning | 1.583 B | 77.18 | **67.89** | 47.95 | 64.34 |
| LoRA | 1.585 B | 75.79 | 65.71 | 49.05 | 63.52 |
| MoDE 1×Uninitialized Expert | 1.979 B | 76.71 | 67.30 | 49.17 | 64.39 |
| MoDE 2×Uninitialized Experts | 2.376 B | 76.87 | 67.39 | 49.47 | 64.58 |
| MoDE 2×Frozen Experts | 2.376 B | 76.99 | 66.94 | 49.07 | 64.33 |
| MoDE 2×Experts | 2.376 B | **77.47** | 67.83 | **49.50** | **64.93** |
|    vs. Full-parameter Fine-tuning | ↑0.793 B | ↑0.29 | ↓0.06 | ↑1.65 | ↑0.59 |
|    vs. LoRA | ↑0.789 B | ↑1.68 | ↑2.12 | ↑0.45 | ↑1.41 |

Table 1: Multi-domain performance comparison of several methods which demonstrates that MoDE provides the best overall performance and capability retention. We adapt each method to the *Math + Coding* dataset, and compare the performance on the test sets for math, coding, and English retention.

**Baselines.** We benchmark MoDE against two widely-used domain adaptation methods: full-parameter fine-tuning (Houlsby et al., 2019) and LoRA (Hu et al., 2021). We exclude similar low-rank methods, such as CoDA (Lei et al., 2023) and QLoRA (Dettmers et al., 2023), which prioritize computational efficiency and exhibit lower accuracy than LoRA. Additionally, we conduct a comprehensive hyperparameter search for LoRA (Table 2) and report results for the best-performing LoRA configuration.

**Chosen MoDE Configuration**. We configure MoDE with three blocks, each containing two transformer layers per expert. For a detailed comparison of different configurations, refer to Table 3.

**Training Details** All methods are trained on a cluster of 128 TPUv3 accelerators across 8 servers. We use a batch size of 128, a learning rate of 0.001, and train for 50k steps.

## 4.2 MULTI-DOMAIN ADAPTATION

**Overall Performance.** We evaluate MoDE, LoRA, and full-parameter fine-tuning on the *Math*, *Code*, and *English* test sets. We train the baseline models directly on the mixed dataset, *Math + Code*. For MoDE, we follow the training procedure described in Section 3.2, where individual MoDE models are first trained on *Math* and *Code* training sets independently, and then composed using the *Math + Code* dataset. As shown in Table 1, MoDE with two experts (one for *Math* and one for *Code*) achieves the best overall performance and retention capabilities. On average, MoDE outperforms full-parameter fine-tuning by 0.59% and LoRA by 1.41%. Notably, MoDE achieves target domain performance comparable to full-parameter fine-tuning, outperforming it by 0.28% on *Math* while trailing by 0.06% on *Code*, and provides a 1.55% improvement in retention on the *English* dataset. Compared to LoRA, MoDE delivers a higher accuracy on the target domains, with a 1.68% improvement on *Math* and 2.12% on *Code*, and surpasses LoRA by 0.45% in *English* retention.

**MoDE Configurations.** We address two questions in exploring different configurations:

1. Which configuration of MoDE modular experts provides the best performance?
2. Which composition strategy yields the best multi-domain performance?

For the first question, we evaluate MoDE with only one uninitialized expert and vary two hyperparameters: the number of MoDE blocks, and the number of transformer layers per expert within each block. As shown in Table 3, increasing the total number of expert layers leads to higher accuracy in the target domain. To balance the number of added parameters, domain-specific accuracy, and capability retention, we configure the MoDE model with three MoDE blocks, each containing two transformer layers per expert (six layers per expert total).

For the second question, we compare against three alternative MoDE configurations. We first measure the benefit of re-using experts by training MoDE configurations with one or two randomly initialized experts: MoDE 1×Uninitialized Expert measures the techniques performance when training a new expert on the multi-domain dataset from scratch, and MoDE 2×Uninitialized Experts accounts for the

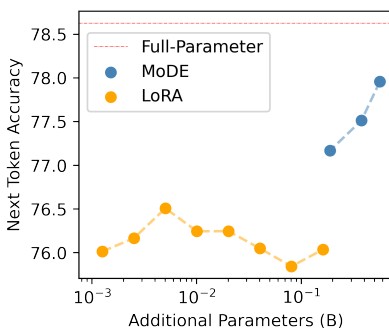 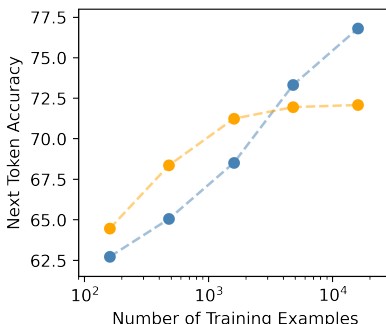

Figure 3: **Scalability of adaptation methods.** We find that MoDE scales better than LoRA as the number of training examples and the number of parameters added increases. In the left figure, we increase the number of adapter parameters for LoRA by increasing the rank and MoDE by increasing the number of expert layers, and find that MoDE provides higher accuracy than LoRA with more trainable parameters. In the right figure, we generate versions of the *Code* with different numbers of training examples, and train a LoRA adapter and a MoDE expert for the same number of training steps on each. Although LoRA provides better accuracy on small datasets up to ∼1k training examples, we find that MoDE's accuracy is better on large datasets, demonstrating that MoDE scales better with more training data than LoRA.

difference in the number of parameters. We also measure how much fine-tuning on the multi-domain dataset benefits accuracy. For this configuration (MoDE 2×Frozen Experts), we train only the gating function and re-use the weights single-domain training: one expert trained on *Math* and the other trained on *Code*. According to Table 1, we find that more experts, initialization, and multi-domain fine-tuning all improve performance.

In addition, we examine how much mixed data from *Math + Code* is required for strong performance. As discussed in Appendix C, freezing the experts during composition yields better training sample efficiency compared to other methods. This finding highlights the potential of our approach for privacy-preserving training, where different entities can independently train domain-specialized experts and later combine them into a coherent model, without sharing expert weights and only sharing sharing a small amount of training data.

## 4.3 SCALABILITY OF MODE

We evaluate the scalability of MoDE along two dimensions: scaling with additional parameters and scaling with increased training examples. We use LoRA as the baseline in both experiments and conduct experiments on a single domain using the *Code* dataset for training and evaluation.

**Scalability with Additional Parameters.** We compare the accuracy improvements of MoDE and LoRA as additional parameters increases. For LoRA, we increase parameters by increasing the rank of the decomposition matrices, ranging from 8 to 2,048. For MoDE, we increase the number of transformer layers per expert while keeping the number of MoDE blocks constant. As shown in Figure 3, while LoRA is parameter-efficient, it struggles to convert additional parameters into higher accuracy. In contrast, MoDE shows a continuous improvement in accuracy as more parameters are added, indicating that MoDE scales more effectively with more parameters.

**Scalability with Training Examples.** We evaluate how well MoDE and LoRA leverage additional training data to improve accuracy. Different versions of the *Code* dataset were created, each containing varying amounts of training examples. Both MoDE and LoRA were trained on these dataset versions, with the number of training steps keep the same across all experiments. As shown in Figure 3, we observe that LoRA performs better on smaller datasets (hundreds to thousands of training examples), but its accuracy plateaus as the number of training examples exceeds one thousand. In contrast, MoDE continues to improve as the number of training examples increases, demonstrating superior scalability with larger training sets.

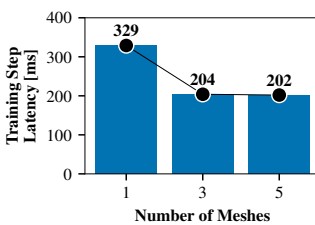 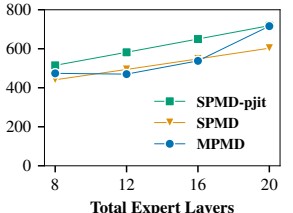 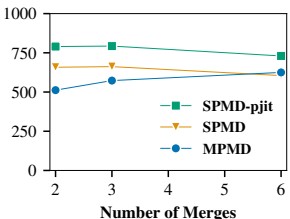

(a) The cost of resharding between meshes is lower than the cost of model parallelism with experts.

(b) MPMD enables faster training when the latency from the expert layers is close to the latency from the backbone.

(c) As the amount of merges increases, the speed of MPMD training decreases due increased communication costs.

Figure 4: **Evaluation of flexible sharding configurations** enabled by the MoDE model architecture.

### 4.4 ACCELERATING TRAINING

Through our experiments, we find several MPMD-enabled sharding configurations which increase training speeds by up to 38% over SPMD model-parallelism. The results confirm that the reduction in communication costs by running backbones and experts on fewer accelerators outweighs the added costs of resharding across meshes when merging the intermediate outputs of models.

All experiments use 8 TPUv5e accelerators which are managed by an orchestration layer that supports MPMD (Anonymized, 2024). We implement MPMD training by extending a distributed machine learning framework (Anonymized, 2024) implemented in Jax (Bradbury et al., 2018). While we use TPUs and a Jax-based framework as our training environment, we underscore that the key idea of using MPMD to increase training speed in distributed settings extends to other accelerators (e.g., GPUs) and training systems such as PyTorch (Paszke et al., 2019). We measure the training performance, but note that many of the performance improvements enabled by MPMD sharding strategies may also benefit model serving. In details, we compare the following configurations:

1. **MPMD** divides the TPUs into different meshes to which backbone or experts are assigned.
2. **SPMD** creates a single mesh which consists of all 8 TPUs, backbone and experts.
3. **SPMD-pjit** uses our training library's to parallelize model training.

**Cost of Resharding.** We compare the following configurations, which train MoDE using 4 experts where each consists of 6 transformer layers divided evenly into two blocks:

1. *1 mesh* uses model parallelism to shard the model across a single mesh.
2. *3 mesh* assigns the backbone to 4 TPUs and each expert shares 2 TPUs with another expert. The backbone and the experts are sharded on their meshes using model parallelism.
3. *5 mesh* assigns the backbone to 4 TPUs, and one expert to each of the remaining TPUs. The backbone is sharded with model parallelism.

We find that the cost of resharding between meshes is lower than the cost of model parallelism (Figure 4a). The reduction in training step latency from 329 ms for *1 mesh* to 204 ms, a 38% reduction, results due to the reduction in communication caused by the MPMD sharding configuration. Because the latency of the *3 mesh* and *5 mesh* configurations is similar, we conclude that the cost to reshard does not increase with the number of meshes.

**Configuring the Expert Size.** We configure a mixture of 2 expert divided into two blocks and change the total number of transformer layers added. For example, setting the expert block size to 2 transformer layers adds 8 total transformer layers to the model. The backbone is assigned to 6 TPUs with a model parallel sharding, and each expert is assigned to one unoccupied TPU.

We find that SPMD performs better than MPMD when the expert size is small or large (Figure 4b) due to *under-utilization* of the TPUs. When the expert size is small (e.g., 8 added layers total) and we train using MPMD, the expert blocks finish executing before the backbone block, so the expert TPUs remain idle until the backbone block completes and the merge begins. Similarly, when the expert size is large (e.g., 20 added layers total), the backbone blocks finish before the expert blocks so the

backbone's TPUs idle until the expert complete. In contrast, SPMD executes the backbone and expert blocks sequentially on all TPUs, ensuring that each TPU is always utilized.

When the runtime of a backbone block is similar to the runtime of the expert blocks, MPMD training results in high utilization of the accelerator. Several settings impact the TPU utilization using MPMD which can be configured to ensure high utilization, such as the number of accelerators allocated to each mesh, the size of the expert blocks, and the sharding strategy on each mesh.

**Cost of Merges.** We train a mixture of 4 experts with 6 transformer layers each. The backbone is assigned to a mesh of 4 TPUs, and each expert is assigned to 1 unoccupied TPU. We change the number of merges which determines the number of blocks (e.g., 3 merges divide the backbone and each expert into three blocks). We find that MPMD training speed decreases as the number of merges increases due to the added communication costs incurred by each merge (Figure 4c).

## 5 CONCLUSIONS

Adapting pre-trained language models (PLMs) to complex, multi-domain settings is increasingly important as these models are deployed in specialized environments requiring diverse capabilities. To address this challenge, we propose Modular Domain Experts (MoDE), a scalable technique for adapting PLMs to multi-domain tasks. MoDE introduces modular, domain-specialized experts while preserving the general knowledge of the PLM by keeping its weights frozen. MoDE allows experts to scale with both the number of parameters and the amount of training data, outperforming standard parameter-efficient methods like LoRA by 1.4% on a challenging multi-domain dataset. Additionally, MoDE delivers competitive performance compared to full-parameter fine-tuning, achieving 1.5% better retention of general capabilities and 0.6% higher accuracy across all subdomains. To further optimize training efficiency, MoDE 's architecture supports flexible sharding strategies through MPMD, resulting in up to 38% faster training compared to standard distributed training methods. The ability to compose and reuse modular experts represents a significant advancement in adapting PLMs to complex domains. We hope that our work inspires further research into building modular, expert-based language models.

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

## A  ABLATION OF LoRA CONFIGURATIONS

In Table 2, we provide an ablation study of different LoRA configurations when adapting to the *Code*. We vary the rank of the low-rank matrices introduced for adaptation from 8 to 2048. While a rank of 32 provides the best performance on the *Code* dataset, we find that a LoRA rank of 16 provides the best balance between adaptation to *Code* and capability retention on the *English* dataset. We also test whether to apply LoRA to the feed-forward networks (FFN) and the token embeddings in addition to the attention matrices. We find that applying LoRA to the FFN slightly improves accuracy on *Code* by 0.27% at the cost of a 0.86% decrease in accuracy on *English* retention.

| Configuration | | | Accuracy | | |
|---|---|---|---|---|---|
| Rank | FFN | Embeddings | Code | English | Average |
| 8 | N | N | 76.02 | **48.76** | 62.39 |
| 16 | N | N | 76.17 | 48.64 | **62.40** |
| 32 | N | N | **76.51** | 46.99 | 61.75 |
| 64 | N | N | 76.25 | 48.25 | 62.25 |
| 128 | N | N | 76.25 | 47.75 | 62 |
| 256 | N | N | 76.05 | 46.62 | 61.34 |
| 512 | N | N | 75.84 | 44.39 | 60.12 |
| 1024 | N | N | 76.04 | 43.91 | 59.97 |
| 2048 | N | N | 66.55 | 32.27 | 49.41 |
| 16 | Y | Y | 76.23 | 47.04 | 61.64 |
| 16 | Y | N | 76.44 | 47.78 | 62.11 |
| 16 | N | Y | 75.81 | 48.03 | 61.92 |
| 16 | N | N | 76.17 | 48.64 | **62.40** |

Table 2: Ablation of LoRA configurations when adapting to the *Code* dataset.

## B  ABLATION OF MoDE CONFIGURATIONS

We investigate how MoDE hyperparameters impact accuracy. In Table 3, we vary the number of blocks in a MoDE model and the number of expert transformer layers per block. When adapting to the *Code* dataset, we find that the number of blocks impacts both domain-specific and retention performance, indicating that tuning the number of blocks impacts model performance. We further find that performance on *Code* increases with the number of expert transformer layers which corresponds to the number of added parameters, but does not seem to significantly impact *English* retention.

| Model Configuration | | Accuracy | | |
|---|---|---|---|---|
| Blocks | Expert Layers | Code | English Retention | Average |
| 2 | 1 | 76.98 | 47.87 | 62.43 |
| 2 | 2 | 77.17 | 47.86 | 62.51 |
| 3 | 1 | 77.26 | 47.49 | 62.38 |
| 3 | 2 | 77.51 | **47.95** | 62.73 |
| 6 | 1 | 77.79 | 47.61 | 62.70 |
| 6 | 2 | **77.96** | 47.65 | **62.80** |

Table 3: Ablation of MoDE model configurations when fine-tuning on a coding dataset. We find that accuracy on the coding dataset increases with the number of expert layers. We select the underlined configuration with 3 blocks and a coding expert with 6 transformer layers to strike a balance between domain-specific accuracy, capability retention, and the number of parameters added.

## C    MULTI-DOMAIN DATA EFFICIENCY

We explore how much multi-domain *Math + Code* data is required for strong performance on the target *Math* and *Code* datasets as well as *English* retention. We follow a similar experimental setup to Section 4.3, but adapt the model to the multi-domain *Math + Code* dataset instead of the single-domain *Code* dataset. A smaller amount of multi-domain data is desirable, as it allows each domain owner to contribute minimal data while ensuring that their experts can collaborate effectively with those from other domains.

We compare against two MoDE variations alongside the baselines. The first variation, Frozen MoDE, freezes experts during composition, while the second, Uninitialized MoDE, skips single-domain training entirely. As shown in Figure 5, Frozen MoDE delivers the best performance when less mixture data is available. However, with more training data, the standard MoDE configuration achieves the highest accuracy across all domains.

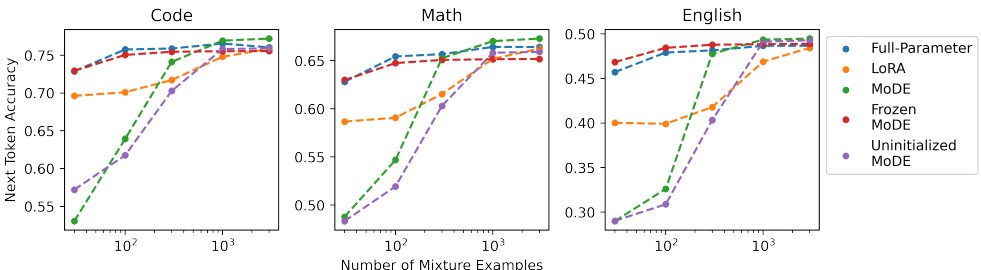

Figure 5: **Efficiency of mixture data.** We examine how much mixture data is required for training. For left to right, we present the next token accuracy on *Code*, *Math*, and *English*. The x-axis for each subplot is the number of examples from *Math + Code* mixture data.

