# OpenReview forum: "Scalable Multi-Domain Adaptation of Language Models using Modular Experts"
_ICLR.cc/2025/Conference — ICLR 2025 Conference Withdrawn Submission_

### Official Review · Reviewer_1GLG · 2024-11-01

**Soundness:** 3
**Presentation:** 3
**Contribution:** 2
**Rating:** 3
**Confidence:** 4

**Summary:**

This paper introduces a mixture-of-experts (MoE) architecture, named Modular Domain Experts (MoDE), to efficiently adapt pre-trained language models to multi-domain tasks. The architecture is designed to optimize the trade-off between domain-specific performance, knowledge retention, and training efficiency. Additionally, the architecture supports integration with sharding strategies for improving computational efficiency.

**Strengths:**

1. The proposed method demonstrates superior performance over existing methods like standard LoRA, excelling in knowledge retention capabilities and offering better scalability as expert parameters and training data increase.
2. The architecture allows for flexible integration with different sharding strategies, such as the MPMD-enabled sharding strategy, effectively accelerating training speeds.

**Weaknesses:**

1. The baselines compared in this work are limited to standard LoRA and SFT. The concept of integrating modular domain expert is somewhat similar to prior papers on MoE and LoRA-MoE architecture (e.g., [1][2][3]), while the difference is MoDE applies transformer block as expert instead of feed-forward networks or LoRA block. The paper may need to add discussions of these differences to clarify its novelty. Furthermore, including other MoE or LoRA-MoE architectures as baselines would help demonstrate the superiority of the proposed approach.

2. Many low-level details are omitted, such as training and test datasets, hyperparameters of baseline methods and the two-stage training procedure. Including these details would help understand the method and facilitate future research endeavors.

3. Although there is no consensus, models >10b parameters are generally considered LLMs. The backbone pretrain model in this paper is only 1.5b, it would be better to try multiple models with larger parameter sizes and apply other PLMs such as Phi and LLaMA to demonstrate scalability.

4. In Table 1, the baseline methods are trained on the mixed dataset, whereas MoDE is trained on Math and Code, respectively, and then on the mixed data. This seems unfair as it has more training steps.

[1] LoRAMoE: Alleviate World Knowledge Forgetting in Large Language Models via MoE-Style Plugin

[2] MOELoRA: An MOE-based Parameter Efficient Fine-Tuning Method for Multi-task Medical Applications

[3] MixLoRA: Enhancing Large Language Models Fine-Tuning with LoRA-based Mixture of Experts

**Questions:**

The effectiveness of MoDE is sensitive to the configuration, yet the criteria for selecting configurations remain confusing. In Table 3,
 *  How do you reach the conclusion that configuring three MoDE blocks, each containing two transformer layers per expert strikes a balance?
* The performance increases with the number of blocks increases, why do you select 3 instead of 6?

**Details Of Ethics Concerns:**

There are no concerns with this submission

---

### Official Review · Reviewer_UPpY · 2024-11-03

**Soundness:** 2
**Presentation:** 2
**Contribution:** 2
**Rating:** 3
**Confidence:** 5

**Summary:**

This paper introduces Modular Domain Experts (MoDE), a mixture-of-expert model that leverages separately trained domain experts. In this architecture, each MoDE expert consists of several transformer layers where experts process tokens in parallel to the backbone model and combined outputs per "blocks". In terms of training, MoDE includes (1) separately training domain experts in their domain data, and (2) fine-tuning all experts with unified data. The paper focuses on maximizing the performance of expert domains while maintaining the general performance of the backbone model in resource-constrained settings. In addition to the performance, the paper proposes parallelizing block execution, enabling speed-up during training.

**Strengths:**

1. Considering the scale of current LLMs and the required cost to train such models, modular mixture-of-expert (MoE) architectures for multi-domain adaptation while maintaining the general-purpose capabilities of the backbone model, as proposed in this paper is quite important.

2. Apart from performance, the paper also considers parallelization during training, which is another important aspects of current LLMs. Although the paper only shows for a particular infrastructure setting and only for training, the proposed parallelization results in an important speed-up.

**Weaknesses:**

1. The paper disregards a large body of work such as [1], [2], [3], [4], [5] that are extremely relevant and similar to the proposed method in many aspects. Some of these works propose parameter-efficient MoE architecture, and others propose modular MoEs where experts are trained separately.

2. Connected to 1st point, the proposed MoDE architecture only compared with full model fine-tuning and simple LoRA methods while there are multiple baselines that should be considered such as BTX [1], BAM [2] or MoE-LoRA methods.

3. For both full model fine-tuning and LoRA, only one variant where the backbone model is fine-tuned using the combined dataset (Code+Math) is presented. Another set of comparisons where the backbone model is fine-tuned per domain separately, and also fine-tuned using combined dataset but including English data as well required.

4. It is unclear why the paper does not include any benchmark on Code, Math, and general language understanding in English. Only next-token prediction per domain is presented however, these results may not correlate with downstream benchmarks in the same degree.

5. For the selected MoDE configuration, the additional number of parameters for only 2 domains is 0.789B which corresponds to approximately 50% of the pretrained parameter count (0.789/1.583). This is quite a high number of parameters when the number of new domains gets higher.

[1] Li, Margaret, et al. "Branch-train-merge: Embarrassingly parallel training of expert language models." arXiv preprint arXiv:2208.03306 (2022).
[2] Zhang, Qizhen, et al. "BAM! Just Like That: Simple and Efficient Parameter Upcycling for Mixture of Experts." arXiv preprint arXiv:2408.08274 (2024).
[3] Zadouri, Ted, et al. "Pushing mixture of experts to the limit: Extremely parameter efficient moe for instruction tuning." arXiv preprint arXiv:2309.05444 (2023).
[4] Wu, Xun, Shaohan Huang, and Furu Wei. "Mixture of lora experts." arXiv preprint arXiv:2404.13628 (2024).
[5] Dou, Shihan, et al. "Loramoe: Revolutionizing mixture of experts for maintaining world knowledge in language model alignment." arXiv preprint arXiv:2312.09979 4.7 (2023).

**Questions:**

1. Are the precise details about the domain-specific datasets such as Code, Math, and English available?
2. Why the parameter count is same for Full-parameter Fine-tuning and LoRA in Table 1?

---

### Official Review · Reviewer_hxps · 2024-11-04

**Soundness:** 2
**Presentation:** 2
**Contribution:** 2
**Rating:** 3
**Confidence:** 4

**Summary:**

Dealing with the multi-domain adaptation problem, this paper proposes a simple idea of combining domain-specialized experts as an MoE model (w/o sparse activation). The backbone model is frozen during training as a shared expert to avoid catastrophic forgetting on original domains. Experiments are conducted on three domains with token classification accuracy as the metric. Results show performance improvements.

**Strengths:**

- Simple idea of initializing specialized experts for specific domains and then merge them for overall performance improvement.
- Efficient implementation of modular expert training.

**Weaknesses:**

My current ratings are based on the following weaknesses, and I am open to hearing from the authors if I'm wrong and re-evaulate the paper.

- Limited catastrophic forgetting verification. Only the *English* corpus is utilized for verifying the effectiveness of dealing with catastrophic forgetting (Code + Math → English). More setups would help understand the effectiveness. e.g., Code → Math + English, Math → Code + English, English → Code + Math, Code + English → Math, Math + English → Code.
- Potential unfair comparisons. Since MoDE is not sparsely activated, there are more available parameters participating in the computation **during inference**. It is unclear whether the performance improvement comes from the parameter scaling or the proposed method.
- Unclear settings of the pretrained 1.5B model and the multi-domain datasets.
    - The baseline performance of the pretrained 1.5B model is missing, and it is not clear how LoRA and full-param SFT would improve the performance. Do you train models from scratch on specific datasets or use a pretrained language model? If the 1.5B model is pretrained already, why not present its performance in Table 1? If the model is pretrained from scratch by your own, why not utilize a smaller pretrained model like LLaMA 3.0B or Phi? Are they too good to diminish the performance improvements?
    - It is not clear whether these datasets are seen during the pre-training stage.
    - Besides, if you utilize open-sourced datasets, please consider disclose details about the datasets. e.g., specific subsets of a open-sourced pre-training corpus.
- The paper proposes a similar idea of ModuleFormer and BTX, but they are not discussed or compared as baselines.
    - Shen, Y., Zhang, Z., Cao, T., Tan, S., Chen, Z., & Gan, C. (2023). Moduleformer: Learning modular large language models from uncurated data. *arXiv preprint arXiv:2306.04640*.
    - Sukhbaatar, S., Golovneva, O., Sharma, V., Xu, H., Lin, X. V., Rozière, B., ... & Li, X. (2024). Branch-Train-MiX: Mixing Expert LLMs into a Mixture-of-Experts LLM. *arXiv preprint arXiv:2403.07816*.
    - Maybe sparse upcycling should also be a baseline: Komatsuzaki, A., Puigcerver, J., Lee-Thorp, J., Ruiz, C. R., Mustafa, B., Ainslie, J., ... & Houlsby, N. (2022). Sparse upcycling: Training mixture-of-experts from dense checkpoints. arXiv preprint arXiv:2212.05055.

**Questions:**

- How are the experts initialized?
- What’s the relation between SPMD vs. tensor parallel, and MPMD vs. tensor parallel + data parallel?
- Typos: Line 016: “a general PLMs” → “general PLMs”

---

### Official Review · Reviewer_EVMb · 2024-11-04

**Soundness:** 2
**Presentation:** 2
**Contribution:** 2
**Rating:** 5
**Confidence:** 3

**Summary:**

This work aims to address domain-specific adaptation with pre-trained language models using a mixture-of-experts architecture that augments a general PLM with modular, domain-specialized experts. These experts are trained independently and composed together via a lightweight training process. For a multi-domain task, experts are modularly composed to enhance the model's performance. A lightweight fine-tuning process updates the experts and the gating function to improve the performance on the target task. Experiemntal results on three tasks shows the effectiveness of the proposed method.

**Strengths:**

1. Using mixture-of-experts for multi-task learning is novel and interesting.

2. Experiemtal results on three datasets as well as the ablation studies show the effectiveness of the proposed approach.

3. The motivation is clear and Figure 1 is helpful to understand the proposed method.

4. Related work are comprehensive to let me get the literature.

**Weaknesses:**

1. The selection probability of the gating function is unclear to me. How does the gating function select experts that represent different tasks when an input from certain task is fed into the model? It is crucial to understand how can the MoE model use the domain-specific knowledge from different experts.

2. Since the MoE will significantly increase the parameter size, e.g., in Table 1. It is unfair to compare the performance of MoE model which has more parameters with those have fewer parameters.

3. The time cost in training a MoE model is very important. Althoght this work uses some model-parallel partitioning to accelerate the training. It is also unclear the time and memory cost within the proposed method.

4. Besides, the analysis of how the relationships between the number of experts and the top-k selection affects the performance is also miss in this work. Which has strong relationship to the efficiency and effectiveness of a MoE approach.

**Questions:**

1. What is the time/memory cost regarding training the MoE model?

2. Have the authors conducted experiments on more tasks, and used more experts?

3. Can we analyze the selection probability for different experts given a certain task's input?

4. How to make a fair comparison with a baseline that has the size parameter size as the MoE model?

---

### Note · Authors · 2024-11-15

**Comment:**

We thank the reviewers for their feedback and have decided to withdraw the paper.

**Withdrawal Confirmation:**

I have read and agree with the venue's withdrawal policy on behalf of myself and my co-authors.